# A Role for Human DNA Polymerase λ in Alternative Lengthening of Telomeres

**DOI:** 10.3390/ijms22052365

**Published:** 2021-02-27

**Authors:** Elisa Mentegari, Federica Bertoletti, Miroslava Kissova, Elisa Zucca, Silvia Galli, Giulia Tagliavini, Anna Garbelli, Antonio Maffia, Silvia Bione, Elena Ferrari, Fabrizio d’Adda di Fagagna, Sofia Francia, Simone Sabbioneda, Liuh-Yow Chen, Joachim Lingner, Valerie Bergoglio, Jean-Sébastien Hoffmann, Ulrich Hübscher, Emmanuele Crespan, Giovanni Maga

**Affiliations:** 1Institute of Molecular Genetics IGM-CNR “Luigi Luca Cavalli-Sforza”, via Abbiategrasso 207, 27100 Pavia, Italy; elisa.mentegari@igm.cnr.it (E.M.); federica.bertoletti@igm.cnr.it (F.B.); miroslava.kissova@ntnu.no (M.K.); elisa.zucca83@gmail.com (E.Z.); silvia.galli@cruk.cam.ac.uk (S.G.); g.tagliavini@sms.ed.ad.uk (G.T.); anna.garbelli@igm.cnr.it (A.G.); antonio.maffia@berkeley.edu (A.M.); bione@igm.cnr.it (S.B.); fabrizio.dadda@ifom.eu (F.d.d.F.); sofia.francia@igm.cnr.it (S.F.); simone.sabbioneda@igm.cnr.it (S.S.); 2Department of Molecular Mechanisms of Disease, University of Zürich-Irchel, Winterthurerstrasse 190, CH-8057 Zürich, Switzerland; elena.ferrari@dmmd.uzh.ch (E.F.); hubscher@uzh.ch (U.H.); 3IFOM-The FIRC Institute of Molecular Oncology, 20139 Milan, Italy; 4Swiss Institute for Experimental Cancer Research (ISREC), School of Life Sciences, Frontiers in Genetics National Center of Competence in Research, Ecole Polytechnique Fédérale de Lausanne (EPFL), Station 19, CH-1015 Lausanne, Switzerland; lyowchen@gate.sinica.edu.tw (L.-Y.C.); joachim.lingner@epfl.ch (J.L.); 5UMR1037 INSERM, Cancer Research Center of Toulouse, 2 Avenue Curien, 31037 Toulouse, France; valerie.bergoglio@inserm.fr; 6Laboratoire d’Excellence Toulouse Cancer (TOUCAN), Laboratoire de Pathologie, Institut Universitaire du Cancer-Toulouse, Oncopole, 1 Avenue Irène-Joliot-Curie, 31059 Toulouse, France; jean-sebastien.hoffmann@inserm.fr

**Keywords:** alternative lengthening of telomeres (ALT), DNA polymerase λ, DNA double-strand breaks (DSBs), extra-chromosomal telomeric repeats (ECTRs), promyelocytic leukemia (PML) bodies, telomere dysfunction-induced foci (TIFs), telomere stress, microhomology-mediated strand transfer (MMST) activity

## Abstract

Telomerase negative cancer cell types use the Alternative Lengthening of Telomeres (ALT) pathway to elongate telomeres ends. Here, we show that silencing human DNA polymerase (Pol λ) in ALT cells represses ALT activity and induces telomeric stress. In addition, replication stress in the absence of Pol λ, strongly affects the survival of ALT cells. In vitro, Pol λ can promote annealing of even a single G-rich telomeric repeat to its complementary strand and use it to prime DNA synthesis. The noncoding telomeric repeat containing RNA TERRA and replication protein A negatively regulate this activity, while the Protection of Telomeres protein 1 (POT1)/TPP1 heterodimer stimulates Pol λ. Pol λ associates with telomeres and colocalizes with TPP1 in cells. In summary, our data suggest a role of Pol λ in the maintenance of telomeres by the ALT mechanism.

## 1. Introduction

The Alternative Lengthening of Telomeres (ALT) is used by telomerase negative cancer cells to avoid shortening of telomeric DNA. The G-rich single strand (ss) 3′-end of telomeric DNA invades the double strand (ds) of another telomeric DNA tract, forming a D-loop structure. DNA synthesis is then initiated at the 3′-end of the annealed G-strand, using the complementary C-rich strand as a template [1].

The ALT process can be initiated by a homologous recombination (HR) mechanism or by an HR-independent pathway [2]. In the HR-dependent mechanism, the ss G-rich overhang binding protein Protection of Telomeres protein 1 POT1, which represses HR, is replaced by replication protein A (RP-A) [3], allowing binding of RAD51, which promotes D-loop formation followed by elongation of the ss G-rich strand using the complementary strand as a template. In the HR-independent mechanism, instead, circular telomeric DNA molecules called extra-chromosomal telomeric repeats (ECTRs) are generated [4,5,6]. In particular, ECTRs constituted by a continuous C-rich telomeric strand paired to a discontinuous (containing nicks or gaps) complementary strand, called c-circles, are specifically generated during ALT [7]. It has been proposed that the G-rich ss 3′-end of a telomere may pair to the complementary strand on the c-circle, providing a template for rolling circle-type DNA synthesis [8,9].

ALT can also result from repair of DNA double-strand breaks (DSBs) at internal telomeric sequences. Telomeres are genomic regions difficult to replicate [10,11,12] and spontaneous DSBs occur frequently at telomeric regions, causing a persistent DNA damage response [11]. In addition, ALT cells are particularly prone to DSBs at telomeres, as a consequence of replication fork stalling and mutations in the ATRX/DAXX chromatin remodeling complex, often present in ALT cells [13,14]. Telomeric DSBs were indeed shown to promote ALT-dependent telomere recombination [15]. Internal DSBs at telomeres can be repaired also by an alternative microhomology-mediated non-homologous end joining pathway (a-NHEJ) [2]. The a-NHEJ dependent on PARP-1 and DNA ligase 3 has been suggested to play a role in the repair of internal telomeric DSBs [16,17]. Moreover, in ALT cells, DSBs can trigger unidirectional break-induced replication (BIR), resulting in telomere elongation [18,19]. Two BIR-dependent pathways have been identified: one RAD52-depedent, which does not lead to c-circle formation, and one RAD52-independent, which relies on Bloom helicase and DNA polymerase (Pol) δ, which is essential for c-circle generation [20] and may be linked to the processing of collapsed replication forks [21].

So far, the identity of the Pols involved in the elongation step of ALT has remained elusive. An alternative replisome complex formed by proliferating cell nuclear antigen, replication factor C and Pol δ, has been shown to operate in the context of BIR-dependent telomere synthesis in ALT cells [19]. Experimental evidence also supports a role of the repair enzyme Pol θ in the a-NHEJ pathway responsible for telomere fusions in the presence of non-functional shelterin complexes, but it is not known whether Pol θ also contributes to repair of internal telomere DSBs in ALT cells [22,23]. Finally, a study suggested a role for the specialized translesion synthesis (TLS) Pol η in promoting replication stress tolerance at ALT telomeres, possibly during HR-dependent replication fork restart or by allowing bypass of replication blocks such as G-quadruplex and oxidative lesions [24]. Whether additional Pols operate in the context of the various ALT pathways is currently not known.

All the pathways proposed to initiate ALT and described above require enzymes able to catalyze DNA strand transfer and annealing reactions followed by DNA synthesis. Human Pol λ is involved in the tolerance of oxidative DNA damage and in the NHEJ repair of DSBs [25] and possesses microhomology-mediated strand annealing and elongation activities on normal or repetitive DNA sequences [26,27]. This microhomology mediated strand transfer (MMST) activity is not sequence-specific, requiring only a minimal homology between the two strands to be joined. Pol λ is also important to overcome DNA replication stress in the U2OS osteosarcoma ALT human cell line. In fact, disabling the S-phase checkpoint together with a silencing of Pol λ results in synthetic lethality in the presence of hydroxyurea (HU) -induced replication fork stalling [28].

Here, we show that silencing human Pol λ in two ALT cell lines strongly reduces ALT-dependent c-circle accumulation and ALT-associated promyelocytic leukemia (PML) bodies formation. Moreover, in the absence of Pol λ, ALT cells, but not telomerase-positive cells, show increased sensitivity to HU-induced replication stress and accumulate DNA damage at telomeres. In vitro, Pol λ promoted annealing of even a single G-rich telomeric repeat to its complementary strand, both on linear and D-loop substrates, and used it to prime DNA synthesis. Telomeric repeats containing RNA (TERRA) and RP-A negatively regulated this activity, while the POT1/TPP1 complex stimulated Pol λ. Pol λ associated with telomeres and colocalized with TPP1 in cells. Our data suggest a previously undetected role of Pol λ in the ALT process.

## 2. Results

### 2.1. Silencing of Human DNA Polymerase λ Specifically Affects the Viability of ALT Cell Lines in the Presence of Replication Stress

Pol λ was silenced in the telomerase-negative cell line Saos-2 (Figure 1A, lanes 3, 4; Appendix A), which uses the ALT mechanism to elongate its telomeric DNA ends. As shown in Figure 1B, silencing of Pol λ alone already significantly reduced cell viability, compared to controls. A similar reduction in viability was observed in control cells after stalling replication forks with HU treatment for 24 h, followed by 24 h of recovery, to allow for replication fork restart, in order to highlight failure in overcoming replication stress (Figure 1B). Combination of HU treatment with Pol λ silencing resulted in an additional loss of viability, suggesting that the effects of HU-induced replication stress were worsened by the absence of Pol λ. In a similar experiment, silencing of Pol λ in U2OS ALT cells (Figure 1A, lanes 5, 6; Appendix A) reduced cell viability similarly to HU treatment. However, combination of Pol λ silencing and HU treatment caused a further reduction in viability (Figure 1C), as observed with Saos-2 cells and according to previous observations [28]. In BJ-hTERT cells, which use telomerase-dependent elongation of the telomeric DNA ends, silencing of Pol λ was effective (Figure 1A, lanes 1, 2) and reduced cell viability (Figure 1D), to a similar extent as HU treatment alone. However, contrary to what has been observed on ALT cell lines, a combination of Pol λ silencing with HU treatment did not result significantly in further loss of viability (Figure 1D).

These results suggested that Pol λ may play a specific role in overcoming DNA replication stress in ALT cells, but not in telomerase-expressing cells.

### 2.2. The Levels of DNA Polymerase λ Correlate with the Cellular ALT Activity

In order to verify whether the ALT process was directly affected by the absence of Pol λ, we monitored the levels of c-circles, a validated marker for ALT activity, in both Saos-2 and U2OS cells. Different amounts of amplified c-circles DNA from Saos-2 (Figure 2A) or U2OS cells (Figure 2C), either wild type or silenced for Pol λ, were spotted on a membrane and hybridized with a ^32^P-labelled telomeric probe. Amplified DNA from telomerase-positive HeLa cells was used as a negative control. Both Pol λ-silenced ALT cell lines (Figure 2A, mid lane, Figure 2C, bottom lane) showed a reduction in c-circle DNA, with respect to the scrambled siRNA-treated cells (Figure 2A, bottom line; Figure 2C, mid lane). Quantification of three independent experiments indicated that Pol λ silencing reduced c-circle formation in Saos-2 cells by 50% (Figure 2B).

In order to investigate whether the ALT activity was directly dependent on Pol λ levels, the amount of c-circles was next compared between U2OS cells wild type and either silenced for Pol λ or overexpressing ectopic c-Myc Pol λ. As expected, silencing of Pol λ reduced c-circles formation with respect to wild type cells (Figure 2D, compare lane 3 with lanes 1, 4), while high levels of Pol λ (assessed by Western Blot in Figure 2E) translated into increased amount of c-circles with respect to wild type cells (Figure 2D, compare lane 2 with lanes 1, 4) thus suggesting that the higher level of Pol λ leads to enhanced ALT activity.

The effect of Pol λ silencing on the formation of ALT-associated PML nuclear bodies (APBs) was tested by fluorescence microscopy analysis of the colocalization of the PML protein with telomeres. APBs are specific markers for ALT activity and are considered sites of telomere DNA synthesis [29]. Cells were sliced into 1µm z-stacks to quantify the co localization in nuclei. Silencing of Pol λ in U2OS cells (Figure 2H; Appendix A) resulted in a highly significant reduction of APBs with respect to control U2OS cells (Figure 2F,G).

Finally, the effects of Pol λ silencing on the cell cycle in ALT cells. As shown in Figure 3A,B, silencing of Pol λ in U2OS cells resulted in an accumulation of cells in late S-G2 phases compared to the scrambled control at the 9 h time point (28.59% vs. 17.67% respectively), with a concomitant reduced number of cells progressing to the G1 phase of the next cycle (36.41% vs. 44.55%, respectively). On the other hand, silencing of Pol λ in BJ-hTERT immortalized fibroblasts, which are telomerase proficient, did not result in a significant delay in the cell cycle (Figure 3C–E). This suggested that silencing of Pol λ resulted in a slower progression through late S-G2 phases, specifically in ALT cells.

Together, these results suggested a direct relationship between Pol λ levels and ALT activity, as measured by c-circles and APBs formation. Since telomeres are replicated in the late-S phase, the observed cell cycle delay upon Pol λ silencing is consistent with a possible role of Pol λ in aiding telomeric ends replication and/or processing.

### 2.3. DNA Polymerase λ Can Catalyze Annealing and Elongation of a G-Rich Telomeric Strand to a Complementary C-Rich Telomeric Template Sequence

ALT-dependent telomere elongation requires that a G-rich ss telomere DNA overhang is annealed to the complementary strand present on either chromosomal telomere DNA or c-circles. This mechanism involves enzymes able to catalyze DNA strand transfer and annealing reactions, followed by DNA synthesis. We have recently shown that Pol λ possesses robust microhomology-mediated strand annealing and elongation activities on normal or repetitive DNA sequences. We termed this activity microhomology mediated strand transfer (MMST) [26].

The ability of Pol λ to perform MMST synthesis of telomeric DNA was then tested with two model substrates, which may be relevant for ALT. First, a 16/48 primer-template (Figure 4A top left scheme) was used, with a dideoxy-terminated primer end, so that no elongation could take place. A ss 25 mer oligonucleotide, containing a 3′-terminal telomeric sequence (TTAGGGTTA) complementary to positions +4 – +12 of the 16/48 template strand (denoted in black in Figure 4A, top), was then supplemented in trans at the beginning of the reaction, which was carried out at 37 °C. Since theoretical Tm for spontaneous annealing of the telomeric 3′-terminal sequence to its complementary template strand is 19.2 °C, at the non-permissive temperature used for the assay, very little annealing should occur, unless stabilized by Pol λ. Addition of Pol λ and all four dNTPs resulted in efficient full-length synthesis (Figure 4A, lanes 1–5). Products of the length expected from the template sequence downstream to the microhomology region appeared only in the presence of the correct nucleotide combinations (Appendix A, lanes 3–8), but not with the 25mer donor alone (Appendix A, lanes 10–14). No nucleotide incorporation was observed in the presence of the dideoxy-terminated 16/48 mer template alone, whose 16 mer primer was 5′-labelled (Appendix A), indicating that DNA synthesis by Pol λ was dependent upon precise annealing of the primer, supplemented in trans, to a microhomology region on the template strand. Moreover, the closely related X-family Pol β could not efficiently replace Pol λ in this reaction, suggesting that this activity was specific for Pol λ (Appendix A).

Next, the 48 mer acceptor DNA template was annealed to a complementary 48 mer oligonucleotide, bearing a stretch of 16 non-complementary nucleotides in the central region, forming a D-loop structure, with an ss “bubble”, encompassing the telomeric -AATCCCAAT- sequence required for the 25 mer ss donor DNA to anneal (Figure 4A, top right scheme). A 25mer ss donor DNA, bearing a complementary telomeric sequence, within the “bubble”, was provided in trans to mimic the G-rich strand invasion proposed to occur in vivo during ALT. Again, Pol λ was able to promote annealing and elongation of the donor 25 mer, reaching the end of the template (Figure 4A, lanes 11–15), indicating strand displacement activity.

### 2.4. Replication Protein A Restricts DNA Polymerase λ Microhomology-Mediated Strand Transfer Activity

RP-A and POT1/TPP1 are important regulators of telomeres homeostasis and ALT [3,30,31,32,33]. We therefore tested whether these proteins affected also the MMST reaction by Pol λ.

First, Pol λ was titrated with the 16/48 and 48/48 substrates, in the presence of the heterotrimeric human RP-A at 1:1 M/M ratio with the DNA substrate. RP-A inhibited the overall DNA synthesis on both substrates (Figure 4A lanes 6–10; 16–20). Titration of both the linear and “bubble” acceptor templates in the presence of RP-A (Figure 4B), showed that RP-A reduced the efficiency of substrate utilization (Vm/Km) by Pol λ on both substrates (Figure 4C and Table 1), suggesting that the effect of RP-A was not dependent on the nature of the DNA template. The inhibitory effect of RP-A could be reduced by increasing the number of target sequences on the template strand. In fact, the efficiency of the MMST DNA synthesis reaction in the presence of RP-A was higher on a 5xtel-16/48 mer acceptor DNA bearing four possible annealing sites (Appendix A, lanes 1–5), than on the 16/48 mer with a single site (Appendix A, lanes 6–10). Overall, the affinity of Pol λ for the 5xtel-16/48 target DNA was increased 2-fold (Table 1).

Thus, RP-A acts as a negative regulator of MMST by Pol λ, likely preventing the annealing between the donor and the template DNA substrates.

### 2.5. The POT1/TPP1 Complex Stimulates the Microhomology-Mediated Strand Transfer Activity of DNA Polymerase λ

Next, the POT1/TPP1 complex was tested in the MMST reaction with Pol λ. As shown in Figure 4D, in the presence of a 5′-labelled 26-mer ss donor DNA strand, bearing the consensus sequence for POT1 binding (5′-TTAGGGTTAG-3′), in combination with a “bubble” 48-mer template (Figure 4D, top scheme), contrary to RP-A, the addition of POT1/TPP1 stimulated MMST primer elongation by low concentrations of Pol λ (Figure 4D, compare lanes 1, 2 with lanes 5, 6, 9, 10). The best stimulation was observed between 15 nM or 30 nM POT1/TPP1 and Pol λ between 30 and 60 nM (Figure 4E, F; Appendix A). Excess of Pol λ (Figure 4D, compare lanes 3, 4 with lanes 7, 8, 11, 12) or of POT1/TPP1 (Figure 4E,F) resulted in less or no stimulation. POT1/TPP1 increased the MMST activity of Pol λ also under conditions were DNA synthesis was limited to the first 6–7 nt (Figure 4G, compare lane 2 with lanes 3–6; Appendix A), that is before full strand displacement (Figure 4G, top scheme). This suggested that POT1/TPP1 increased the ability of the Pol λ-primer complex to find its target template, rather than its strand displacement activity. To directly test this hypothesis, the 48/48mer bubble template was titrated in the reaction under the same limiting conditions. As shown in Figure 4H and Appendix A), POT1/TPP1 increased the DNA substrate utilization efficiency by Pol λ (Figure 4I).

Thus, contrary to RP-A, the POT1/TPP1 complex positively influenced the MMST activity of Pol λ.

### 2.6. TERRA RNA Can Function as a Substrate for the Microhomology-Mediated Strand Transfer Activity by DNA Polymerase λ

The long noncoding RNA TERRA has been shown to form RNA/DNA hybrid structures (R-loops) at telomeres, especially in ALT cells, where TERRA levels are usually higher than in telomerase-positive cells. These R-loops have been proposed to function also as starting points for ALT DNA synthesis [34]. R-loops are also strong DNA replication blocks, whose bypass may involve replication fork restart [12]. Since we have previously shown that Pol λ has the unique ability to elongate an RNA primer annealed to a DNA template [35], we tested whether Pol λ was able to promote MMST DNA synthesis on our 16/48 mer template, starting from an RNA primer containing a telomeric repeat sequence at its 3′-end (26mer TERRA). As shown in Figure 5A, Pol λ was indeed able to promote annealing and elongation of the 26mer TERRA primer, but to a much lower extent than its equivalent DNA primer (Figure 5A, compare lanes 2–6, with lanes 9–14). A similar drop in efficiency was observed also on the 48/48mer bubble template (Figure 5B, compare lanes 2–3 with lanes 10–11), where most of the TERRA-primed synthesis stopped immediately before the ss/ds junction (product +4, Figure 5B top scheme), while full length products were synthesized when the 26 mer DNA primer was used.

### 2.7. The POT1/TPP1 Complex Favors the Microhomology-Mediated Strand Transfer Activity by DNA Polymerase λ Starting from Telomeric DNA over TERRA RNA

Since TERRA RNA can be used as a primer by Pol λ, we next asked whether and to what extent its presence could interfere with the DNA-primed MMST DNA synthesis. To this aim, competition experiments were carried out on the 48/48mer bubble template, either in the presence of fixed-labeled 26mer DNA and increasing amounts of unlabeled 26mer TERRA (Figure 5B, lanes 4–8), or with a fixed labeled 26mer TERRA primer and increasing amounts of unlabeled 26mer DNA (Figure 5B, lanes 12–16). Both unlabeled competitors (either RNA or DNA) were able to reduce MMST synthesis by Pol λ (Figure 5B, compare lanes 2, 3 and 10, 11 with lanes 4–8 and 12–16, respectively) with similar affinities (Figure 5C). The POT1/TPP1 complex was able to increase the rate of the DNA-primed MMST reaction by Pol λ. Consistently, the presence of POT1/TPP1 reduced the ability of TERRA to inhibit the DNA-primer MMST reaction (Figure 5D, compare lanes 2–6, with lanes 7–21) in a dose-dependent manner (Figure 5E).

Colocalization of Pol λ and TERRA was also attempted in Saos-2 cells. Appendix A A shows an example of colocalization of Pol λ foci (as visualized by α-cMyc Abs) and TERRA foci (as visualized by TERRA-specific FISH). The TERRA foci were sensitive to RNase A treatment, confirming their RNA nature (Appendix A). However, colocalization events were very rare and analysis on a representative number of fields failed to show statistical significance (Appendix A). This might reflect a very transient nature of the interaction between Pol λ and TERRA, making it difficult to capture a sufficient number of events in living cells.

In summary, POT1/TPP1 favors DNA-primed MMST DNA synthesis by Pol λ, effectively counteracting its inhibition by TERRA.

### 2.8. DNA Polymerase λ Associates with Telomeres and TPP1 in ALT Cells

The interaction of Pol λ with telomeric DNA was investigated by immunofluorescence analysis and confocal microscopy on Saos-2 cells, stably expressing ectopic Myc-tagged Pol λ and on U2OS cells. Saos-2 cells were synchronized in S-phase by a low dose of HU and then released for 6 h before analysis, in order to enrich the population of late S-phase cells. Figure 6A shows an example of colocalization of telomeric DNA, detected by telomere-specific FISH, and Pol λ foci, detected by α-Myc Abs. A total of 62 cells were analyzed, corresponding to 1254 Pol λ foci and 4830 telomeric-positive foci, respectively (Figure 6B). Since we observed only between 1 to 3 colocalization events per cell (Figure 6C), we performed a statistical analysis comparing the number of colocalization events observed vs. those predicted to occur under the hypothesis of a random distribution of foci, which indicated that the colocalization was statistically significant (Figure 3D).

In order to confirm that Pol λ and TPP1 colocalize in ALT cells, colocalization between Pol λ (visualized by α-cMyc Abs) and TPP1 (visualized by α-TPP1 Abs) was performed on Saos-2 cells expressing Myc-Pol λ, synchronized with HU and released for 6 h (Figure 7A). A total of 1243 TPP1 foci (range 5–203/cell) and 349 Pol λ foci (range 11–40/cell) were detected (Figure 7B). Since we observed only between 1 to 5 colocalization events per cell (Figure 7C), we performed a statistical analysis comparing the number of colocalization events observed vs. those predicted to occur under the hypothesis of a random distribution of foci, which indicated that the colocalization was statistically significant (Figure 7D). These results indicated that only a fraction of Myc-Pol λ colocalized with TPP1. This can be explained by the fact that Pol λ acts also on non-telomeric chromosomal DNA and the interaction of Pol λ and TPP1 likely occurs only in the subset of TPP1 foci, which are engaged in ALT.

Moreover, the same analysis was performed on endogenous Pol λ in unsynchronized U2OS cells. Figure 7E shows an example of endogenous Pol λ and TPP1 colocalizations. A total of 3461 TPP1 foci (range 29–42/cell) and 2550 Pol λ foci (range 1–103/cell) were detected (Figure 7F). In this case, we observed a higher number of colocalization events per cell (Figure 7G). Once again, statistical analysis under random distribution hypothesis revealed effective colocalization of the two factors (Figure 7H).

All together, these experiments indicated that Pol λ can associate with telomeric DNA in ALT cells.

### 2.9. DNA Polymerase λ Silencing Induces Telomere Stress Specifically in ALT Cells

Telomere dysfunctions can induce DNA damage at telomeres, which results in the appearance of foci of DNA damage response proteins such as P53BP1 at telomeres. Thus, we investigated whether silencing of Pol λ in ALT cells was sufficient to induce such telomere dysfunction-induced foci (TIFs). To this aim, colocalization of TPP1 with P53BP1 was analyzed by confocal microscopy in Saos-2 and U2OS cells either wt or treated with Pol λ-specific siRNA. Figure 8A,C show examples of colocalizations between P53BP1 and TPP1 respectively in Saos-2 and U2OS cells silenced for Pol λ (Pol λ KD). In both ALT cell lines, as shown in Figure 8B,D, Pol λ deficiency results in a statistically significant higher number of TIFs containing both P53BP1 and TPP1, with respect to wt cells. When a similar analysis was performed on BJ-hTERT cells, which use telomerase-dependent elongation of the telomeric DNA ends, no statistically significant difference between wild type and Pol λ KD cells was observed (Figure 8E).

## 3. Discussion

ALT is exploited for telomere maintenance in a wide range of human cancers [36] and is particularly frequent in osteosarcoma and glioblastomas [37,38]. All the proposed mechanisms for ALT initiation involve annealing and elongation by a Pol of a G-rich telomere strand to a complementary C-rich template, often in the context of D-loop structure [1,39]. However, which and how many Pols are involved in ALT synthesis is still imperfectly known.

Here, we have shown that Pol λ has an intrinsic ability to promote MMST synthesis within a D-loop template, bearing C-rich telomere repeats, mimicking the G-rich strand invasion known to occur during ALT. In vivo, Pol λ colocalizes with TPP1 and the POT1/TPP1 complex stimulates Pol λ activity in vitro. Conversely, RP-A and TERRA negatively regulate this reaction. The strand displacement activity of Pol λ observed even in our minimal in vitro system, might be relevant in promoting rolling circle replication through the ds regions of a c-circle template. RP-A has been shown to prevent G-rich ss 3′ overhang elongation in ALT cells [40]. One hypothesized mechanism was that RP-A prevented annealing of the 3′ ss overhang to the ECTRs, thus impeding rolling circle amplification. Our in vitro data showing repression by RP-A of the initial annealing step of MMST synthesis mediated by Pol λ on both linear and D-loop templates, fully support this hypothesis. The c-circles are believed to represent intermediates of rolling circle-type amplification of telomeres, likely formed at the levels of APBs and their levels decline rapidly when ALT is inhibited [7]. Consistently, we have shown that Pol λ silencing reduces c-circle levels and APBs formation, while its overexpression increases c-circles formation. In vivo, replacement of POT1/TPP1 with RP-A at G-rich overhangs of telomeres is favored by TERRA and promotes the HR-dependent ALT mechanism [3,31,34]. Conversely, rolling circle replication starting from c-circles is HR-independent [7]. Thus, our results showing a direct correlation between Pol λ levels and the accumulation of c-circles might suggest a role for Pol λ in this HR-independent ALT mechanism.

Telomeres in ALT cells are particularly prone to replication stress and DSBs formation [10,11]. Repair of DSBs at internal telomeric sequences has been shown to involve a microhomology-mediated a-NHEJ mechanism, which starts with 5′-end resection of the broken ends, to create ss overhangs [17]. The resulting ss G-strand is bound by the POT1/TPP1 complex, which prevents further resection. The a-NHEJ represses ALT by repairing the break before it can be used for strand invasion. However, the ss G-strand bound by the POT1/TPP1 complex formed during the process could also be used to initiate ALT by invasion of another telomere or by annealing to ECTR molecules such as c-circles. We observed that Pol λ depletion increases P53BP1 foci at telomeres in ALT cells, suggesting Pol λ participation in DSBs repair at telomeres. Given that Pol λ has been shown to participate in microhomology-mediated NHEJ [26], its ability to interact with POT1/TPP1 might also promote ALT initiation as a consequence of aberrant a-NHEJ of internal DSBs at telomeres. Interestingly, Pol λ has been shown to be upregulated in many cancer types, including thyroid, ovary and lung carcinomas [41]. These malignancies have been often found to lack telomerase, thus Pol λ overexpression might increase aberrant repair events, leading to ALT initiation.

The ability of Pol λ to promote annealing and elongation of a DNA strand within a D-loop structure, could also suggest additional roles of this enzyme in telomere replication. Telomeres are regions difficult to replicate, due to their repetitive sequences and potential secondary structures, which can lead to replication fork stalling and replication stress. Indeed, D-loop-type structures are very frequent at telomeres in ALT cells, due to the high levels of TERRA [13,34]. Annealing of TERRA at telomeric DNA generates R-loops that are strong blocks for a DNA replication fork and need to be bypassed [12]. Based on our findings, it is possible to hypothesize that, upon fork stalling at the R-loop, Pol λ could use the 3′-OH end of the TERRA RNA to prime DNA synthesis, resulting in a short tract of DNA, which could be used for repriming by the replication fork. Upon degradation of TERRA, known to occur during S-phase, a gap will be left behind the advancing replisome, to be filled by postreplicational DNA repair (possibly by Pol λ itself). Alternatively, a model could be envisaged, where, upon fork collapsing, the 3′-end of the leading strand will re-anneal at a microhomology region downstream the TERRA RNA. Pol λ and the POT1/TPP1 complex could favor this process, allowing Pol λ itself initiating a synthesis at the leading strand. Indeed, three lines of evidence stemming from our work support a role for Pol λ in counteracting replication stress: (i) the increased sensitivity to HU-induced fork stalling observed in ALT cells, when combined with silencing of Pol λ (this work and [28]); (ii) the delay in progression through late-S/G2 phase in Pol λ silenced ALT cells; and (iii) the induction of DNA damage foci at telomeres upon Pol λ silencing in ALT cells.

Depletion of the specialized TLS enzyme Pol η in ALT cells has been shown to induce elevated levels of ALT activity and sister chromatic exchanges between telomeric ends [24]. It has been proposed that Pol η works in an HR-dependent mechanism for restarting the stalled replication fork, or allows bypass of replication blocks such as G-quadruplexes or oxidative lesions, thereby preventing DSB formation and BIR-dependent ALT. On the other hand, Pol η did not seem to play a role in DSBs repair at telomeres. Our data showed that depletion of Pol λ has the opposite effect, namely to repress ALT, while its overexpression increases ALT activity. Thus, it seems that Pol η and Pol λ involvement in replication stress tolerance at telomeres in ALT cells has different outcomes. While Pol η promotes a general bypass of replication blocks avoiding DSB formation, Pol λ could be involved in more specific events, such as R-loops bypass or a-NHEJ DSBs repair, which could also prime ALT synthesis, especially in the presence of elevated levels of Pol λ.

## 4. Conclusions

In summary, the data presented in this work unveil a previously unnoticed link between Pol λ levels and ALT activity in tumor cells. ALT-positive tumors have been associated with higher invasively and poorer prognosis [42,43,44]. Thus, our data suggest a possible exploitation of Pol λ as a potential anticancer target for ALT-positive tumors.

## 5. Materials and Methods

### 5.1. Chemicals

[γ-^32^P]ATP (6000 Ci/mmol) was from GE Healthcare Biosciences; unlabeled dNTPs were from Roche Molecular Biochemicals. All other reagents were of analytical grade and were purchased from Merck or Fluka.

Nucleic acids substrates. The sequences are:
Acceptor linear (16ddC/48mer)5′-CAGTCGATCGATCGA**c** (c = ddC)3′-GTCAGCTAGCTAGCTGCCAATCCCAATCCCAACTATAGAGCGCTGTCA-5′Acceptor bubble (48mer_comp_bub/48mer)5′-CAGTCGATCGATCGAC**AAAACAAAAACAAAAA**GATATCTCGCGACAGT-3′3′-GTCAGCTAGCTAGCTGCCAATCCCAATCCCAACTATAGAGCGCTGTCA-5′5xtel16ddC/48mer (16ddC/5xtel48mer)5′-CAGTCGATCGATCGA**c** (c = ddC)3′-GTCAGCTAGCTAGCTGCCAATCCCAATCCCAATCCCAATCCCAATCCC-5′5xtel acceptor bubble (5xtel48mer_comp_bub/5xtel48mer)5′-CAGTCGATCGATCGAC**AAAACAAAAACAAAAAA**GGGTTAGGGTTAGGG-3′3′-GTCAGCTAGCTAGCTGCCAATCCCAATCCCAATCCCAATCCCAATCCC-5′25 mer donor5′-GTCAGCTAGCTAGCTATTAGGGTTA-3′26 mer donor5′-GTCAGCTAGCTAGCTATTAGGGTTAG-3′26 mer TERRA5′-GUCAGCUAGCUAGCUAUUAGGGUUAG-3′

All substrate oligonucleotides and their corresponding primers were purified on a 15%(*w*/*v*) polyacrylamide, 7 M urea, 30% formamide gel. After elution and ethanol precipitation, their concentrations were determined spectrophotometrically. Bold letters indicate the non-complementary regions introduced to obtain the “bubble” structure. Underlined sequences indicate the microhomology regions available for annealing. Telomeric repeats are shaded in grey. The 25-, 29- 32- and 35-mer oligonucleotides were 5′ labelled with T4 polynucleotide kinase (New England Biolabs) in the presence of [γ-^32^P]ATP. The 26-mer donor oligonucleotide was synthesized with a fluorescent group (6-FAM) at its 5′-end. Each labelled primer was mixed to the complementary template oligonucleotide at 1:1 (*M*/*M*) ratio in the presence of 25 mM Tris-HCl (pH 8.0) and 50 mM KCl, heated at 80 °C for 3 min and then slowly cooled down at room temperature.

### 5.2. Proteins Production and Purification

Recombinant His-tagged human wild-type Pol λ, and human RP-A, were expressed and purified as described [45]. Human Pol β was from TREVIGEN. After purification, the proteins were >90% homogenous, as judged by SDS–PAGE and Coomassie staining. The cDNA of POT1 and TPP1 genes were cloned into pEAK8 vector for mammalian expression of flag-tagged POT1 and non-tagged TPP1 driven by the EF1-a promoter. Protein expression was carried out by transient transfection of both POT1 and TPP1 expression plasmids on a 1:1 ratio into HEK293E cells for 72 h as described [46]. The cells were then harvested and lysed in NETN buffer (40 mM Tris-HCl, pH 8.0, 100 mM NaCl, 1 mM EDTA, 0.5% NP40) supplemented with a protease inhibitor cocktail (Roche) and centrifuged at 15,000g for 20 min. After centrifugation, anti-flag M2 affinity gel (Sigma) was added to the supernatant and incubated at °C for 4 h, followed by washes with NETN buffer and PBS. The POT1/TPP1 complex was eluted into PBS plus 10% glycerol by TEV protease cleavage at 4 °C for 5 h, and TEV protease was subsequently removed by Ni-NTA agarose (Qiagen). The activity of the purified POT1/TPP1 complex was determined by its ability of binding to the telomeric G-stranded oligo (TTAGGG)_3_ and to stimulate telomerase activity in vitro [46].

### 5.3. Cell Lines and Treatments

Saos-2 and U2OS (osteosarcoma) cell line and BJ-hTERT (hTERT-immortalized fibroblasts) were grown in Dulbecco modifies Eagle medium (DMEM) complemented with 10% fetal bovine serum at 37 °C and 5% CO_2_. Pol λ over-expressing Saos-2 clones were grown in DMEM complemented with 10% fetal bovine serum and 500 µg/mL of G418 (Sigma Aldrich) at 37 °C and 5% CO_2_. Hydroxyurea was purchased by Sigma Aldrich. Approximately 5 × 10^3^ U2OS and Saos-2 cells and 2.5 × 10^3^ BJ-TERT cells were seeded in 96-well plate and treated for 24 h at 37 °C with hydroxyurea at the indicated concentrations, then the cell viability was recorded following 24 h of recovery. To synchronize cells for immunofluorescence, clones were treated with 1mM HU for 24 h and fixed in 4% PFA after 6 h of recovery, during the S-phase. Approximately 10^6^ U2OS and Saos-2 cells were seeded in two 10 cm^2^ plates for the c-circle assay in order to perform DNA extraction.

### 5.4. Protein Extracts

To prepare whole-cell extracts, cells were re-suspended into one packed cell volume of buffer containing 10 mM Tris-HCl (pH 7.8), 200 mM KCl, 1× protease inhibitor cocktail (Sigma Aldrich) and 1 mM PMSF. Two packed cell volumes of buffer containing 10 mM Tris-HCl (pH 7.8), 600 mM KCl, 40% glycerol, 0.1 mM EDTA and 0.2% Nonidet P-40 was then added and mixed thoroughly before rocking the cell suspension for 2 h at 4 °C. The cell lysate was then centrifuged at 15,000 rpm at 4 °C for 20 min and the supernatant was collected, aliquoted and stored at −80 °C.

### 5.5. Cytotoxicity Assay

The CellTiter 96 Aqueous One Solution cell proliferation assay (Promega) was used as a colorimetric method for determining the toxicity induced during treatments. Briefly, the day before treatment cells were seeded into 96 well-plates (density 1000–10,000 cells/well). To evaluate the treatment effects, the medium was replaced with 100 µL of fresh complete medium and 20 µL of cell titer reagent; then, after incubation at 37 °C for at least 3 h, the absorbance at 492 nm was recorded.

### 5.6. RNA Interference

Cells were transfected using Lipofectamine RNAiMAX reagent (Life Technologies) according to the manufacturer’s protocol. To target Pol λ transcript it was used either the siRNA Selective Validated signal 26198 (si98) 5′-CAAAAGUACUUGCAAAGAUTT-3′ and Silencer Select Negative Control siRNA #1 (scrambled control) from Ambion, or the siRNA-SMARTpool (M-008746-00-0005, siGENOME human POLL (27343) from Darmacon, comprising four different validated siRNAs with the sequences: 5′-GGGCAGAACUCUUUGAGAA-3′; 5′-GGAGGAGGCUACAGAGAUU-3′; 5′-GGAAGCGGAUGGCUGAGAA-3′; 5′-GACCAAGACUGCCCAGAUG-3′. si98 and SMARTpool gave similar Pol λ silencing efficiencies.

### 5.7. Pol λ over Expression Stable Cell Lines

Saos-2 and U2OS cells were grown on 10 cm dishes to 80% confluence and then the cells were transfected with Lipofectamine RNAiMAX transfection reagent (Life Technologies) following the manufacturer’s protocol in the presence of 2 µg of purified plasmid, pcDNA3-polλ-Myc or the empty vector. Following 24 h after transfection, cells were split to the optimal plating density of a 3 × 10^5^ cells/dish and incubated further for 24 h before the start of the selection process. The optimized selective concentration of G418 was 1 mg/mL. The obtained clones were grown in medium containing 500 µg/mL G418 and the Pol λ expression was checked by Western blot and immunofluorescence assay to select the most proficient clones. The clone Saos λ13 displayed the highest Pol λ expression, while Saos EV was the empty vector control.

### 5.8. Immunofluorescence

#### 5.8.1. Pol λ and TPP1 Colocalization

Saos-2. 10^5^ cells/mL were seeded on glass coverslips 24 h prior to the synchronization with 1 mM HU for 24 h, followed by the 6 h recovery before the subsequent 10mM HU treatment for 24 h. Primary antibodies were anti-PTOP mouse monoclonal antibody (ab57595, Abcam) and anti-Myc tag rabbit polyclonal antibody (ab9106, Abcam); secondary antibodies were anti-rabbit Alexa Fluor 555 and anti-mouse Alexa Fluor 488 (Life Technologies). Nuclei were stained by ProLong Gold solution (Life Technologies). The medium was removed and the coverslips were gently washed once in PBS for 5′ in agitation. Extraction was performed with CSK buffer (10 mM Hepes Buffer pH 7.4, 100 mM NaCl, 300 mM sucrose, 3 mM MgCl_2_) + 0.5% Triton for 2′, followed by a 5 min-wash in CSK buffer in agitation. Cells were fixed in 4% PFA in PBS for 10′ (Paraformaldehyde 16% solution Electron Microscopy Science) on ice. Cell permeabilization was achieved through PBS Triton 0.1% for 3 min. After 3 washes in PBS, the coverslips were stored in 3% BSA in PBST 0.1% at 4 °C until staining. The coverslips were incubated with primary antibody for 2h at RT in humid chamber: α-Myc tag (rabbit) 1:250 in 3% BSA with PBS Tween 0.1% and α-PTOP (mouse) 1:50 in 3% BSA with PBS Tween 0.1%. After washing in PBS, the coverslips were incubated for 1 h RT in humid chamber with the secondary antibodies α-rabbit (donkey) 1:5000 and α-mouse (goat) 1:1000 in 3% BSA with PBS Tween 0.1%, washed in PBS in dark and mounted on slides with Prolong Gold antifade reagent with DAPI (Invitrogen).

U2OS. 3 × 10^5^ cells were seeded on glass coverslips for 24 h. Primary antibodies were anti-ACD (TPP1) rabbit polyclonal antibody (ab220740, Abcam) and anti-Pol λ mouse monoclonal antibody (sc-373844, Santa-Cruz); secondary antibodies were anti-rabbit Alexa Fluor 488 (ab150077, Abcam) and anti-mouse Alexa Fluor 555 (ab150114, Abcam). Nuclei were stained with DAPI (1:250,000 in PBS). Extraction was performed with CSK buffer (10 mM Hepes Buffer pH 7.4, 100 mM NaCl, 300 mM sucrose, 3 mM MgCl_2_) + 0.5% Triton for 2′, followed by a 5 min-wash in CSK buffer in agitation. Cells were fixed in 4% PFA in PBS for 10′ (Paraformaldehyde 16% solution Electron Microscopy Science) on ice. Cell permeabilization was achieved through PBS Triton 0.1% for 3 min. After 3 washes in PBS, the coverslips were stored in 3% BSA in PBST 0.1% at 4 °C until staining. The coverslips were incubated with primary antibody for 2 h at RT in a humid chamber: α-TPP1 (rabbit) 1:100 in 3% BSA with PBS Tween 0.1% and α-Pol λ (mouse) 1:500 in 3% BSA with PBS Tween 0.1%. After washing in PBS, the coverslips were incubated for 1 h RT in a humid chamber with the secondary antibodies α-rabbit (goat) 1:2000 and α-mouse (donkey) 1:1500 in 3% BSA with PBS Tween 0.1%. Following two washes for 5′ in 3% BSA in 1× PBS and one wash in 1× PBS for 5′, coverslips are incubated with DAPI (1:250,000) at RT for 10′. Finally, coverslips are washed twice in 1× PBS and mounted with mounting solution (Aqua-Poly/Mount, Polysciences) and stored at 4 °C in dark. Pol λ and TPP1 foci, colocalizations and nuclei were counted and quantified using ComDet (Fiji) plugin [47]. The best-fit lower threshold was determined using the threshold tool and visually inspected. Images were acquired using a 63× objective on confocal microscope (Zeiss). Cell nuclei were visualized at 405 nm (DAPI) and foci at 555 nm (TRITC) or 488 nm (FITC).

#### 5.8.2. TPP1 and P53BP1 Colocalization

For Saos-2 cells: 75 × 10^4^ SAOS-2 cells were seeded in 10 mm dishes and transfected with 100 pmol of siRNA C- or 100 pmol of siRNA POLL through the use of lipofectamine RNAiMAX (Invitrogen) for 120 h. Cells were harvested at 85–90% of confluence and utilized to validate silencing through Western Blot analysis. At 120 h of silencing coverslips were moved to 24-well plates and washed once in filtered 1× PBS for 5′ at 4 °C. Extraction of soluble proteins is performed using CSK buffer + 0.5% Triton for 2′. After a washing for 5′ in CSK buffer, cells are fixed in 4% PFA. Cells are subsequently permeabilized using 1× PBS + Triton 0.1% for 3′ on ice and washed 3 times for 5′ in 1× PBS. Blocking is carried out in 3% BSA in 1× PBS Tween 0.1% at 4 °C O/N. After 3 washes in 1× PBS1 for 5′ in ice, coverslips were incubated with primary antibodies for 1 h at RT in humid chamber. α-53BP1 (goat, Bethyl, A303-906A) 1:2000 and α-ACD (TPP1 Abcam, ab220740, rabbit) 1:100 in 3% BSA in 1× PBS Tween 0.1% were utilized. After 3 washes for 5′ in 3% BSA in 1× PBS on ice, coverslips were incubated with secondary antibodies for 1 h at RT in humid chamber. α-goat Alexa 555 (Invitrogen, A21432) (1:800) and α-rabbit Alexa 488 (Abcam, ab150077) (1:2000) in 3% BSA in 1× PBS Tween 0.1%. were utilized. Following two washes for 5′ in 3% BSA in 1× PBS and one wash in 1× PBS for 5′, coverslips are incubated with DAPI (1:250,000) at RT for 10′. Finally, coverslips are washed twice in 1× PBS and mounted with mounting solution (Aqua-Poly/Mount, Polysciences) and stored at 4 °C in the dark. TPP1-P53BP1 colocalization events were quantified using JACoP (Just Another Colocalization Plug-in) ImageJ colocalization tool [48]. M2 Manders’ overlap coefficient estimates the amount of colocalizing signal from one channel (green, TPP1) over the other (red, P53BP1) [49,50]. The best-fit lower threshold was determined using the threshold tool and was viusully inpected.

For U2OS cells: 10^5^ cells/mL were seeded on glass coverslips 24 h prior to the synchronization with 1 mM HU for 24h, followed by the 6 h recovery. Proteins were detected with anti-PTOP mouse monoclonal antibody (ab57595, Abcam) at 1:50 dilution and with α-53BP1 (goat, Bethyl, A303-906A) at 1:2000 dilution. The secondary Abs used were α-goat (donkey) Alexa Fluor 488 at 1:400 + α-mouse 555 (donkey) at 1:300 dilution, respectively. Protein colocalizations were analysed using the software ImageJ and CellProfiler [48].

Images were acquired using a 60× objective on confocal microscope (Zeiss). Cell nuclei were visualized at 405 nm (DAPI) and foci at 555 nm (TRITC) or 488 nm (FITC).

#### 5.8.3. Immunofluorescence-FISH and APBs Detection

Saos-2 λ13 cells were seeded on glass coverslips 24 h prior to the synchronisation with 1mM HU for 24 h, followed by 6 h recovery. U2OS cells were transfected by siRNA C- or siRNA POLL; 24 h after transfection, cells were seeded on coverslip and fixed 24 h later.

Cells on coverslips were fixed in 4% PFA for 15 min at room temperature, washed 2 times in PBS, permeabilized in a 0.2% triton solution for 10 min at RT and saturated in PBS, BSA 3% solution for 30 min.

PML immunostaining: PML detection was performed by the mouse monoclonal α-PML antibody (SantaCruz SC966) diluted 1/250 in PBS, 1% BSA, 0.1% triton for 2 h at RT followed by 3 washes in PBS and incubation with an alexa-488 goat anti mouse secondary antibody for 1 h at RT. After incubation with the secondary antibody, cells were washed in PBS, incubated in 2% PFA for 5 min at RT and washed again.

For telomeres FISH detection, cells were treated with 100 µg/mL RNase A in SCC2X, for 1 h at 37 °C in dark. After 2 washes in SSC 2×, samples are treated with increasing ethanol concentration: 70% EtOH, 85% EtOH, 100% EtOH. For the hybridization step, cells are incubated with PNA (Peptide Nucleic Acid) Telomere C-rich Cy3-labelled probe 67µg/mL (MIX Q^PCR^Panagene) diluted 1:80 in hybridization buffer at 80 °C for 10′, for DNA denaturation, and at 30 °C O/N for probe hybridization with telomeric DNA sequences. Coverslips are washed in SSC 0.4× + 0.3% NP-40 at 68 °C for 3′ and SSC 2X + 0.1% NP-40 at RT for 1′. Nuclear DNA is stained with DAPI solution 1:5000 in PBS for 5′ in dark. Coverslips are mounted on slides with fluorescent mounting medium (Dako) and stored at 4 °C.

Cells were visualized at the confocal microscope (Zeiss) with 488 nm laser for FITC (Myc tag), 561 nm laser for Cy3 (for telomeric probe) and 405 nm for DAPI.

### 5.9. Terra RNA FISH

Cells were seeded on coverslips 12 h before immunostainings. To prevent RNA degradation ribonucleoside-vanadyl complex—an inhibitor of various ribonucleases—was added to all the solutions and buffers used to a final concentration of 10 mM, except for the negative control slides treated with RNaseA. Cells were fixed and permeabilized with 3.6% formaldehyde; 0.3% Triton X-100; 1× PBS for 10 min and washed 3 times with 1× PBS for 5 min/wash. For the negative control, slides were treated with 0.1 mg/mL RNAseA for 1 h at 37 °C in a humid chamber and washed 3 times with 1× PBS for 5 min/wash. Cells were dehydrated with 70% ethanol, 95% ethanol and kept overnight at −20 °C in 100% ethanol. Next day, the cells were air-dried and incubated with pre-denatured telomeric C-rich PNA probes coupled with FITC (Panagene # F1001). The probe was diluted in hybridization buffer to the final concentration of 50 mM and denatured at 85 °C for 10 min (it is important to denature only the probes not in contact with the cells to avoid unspecific binding with denatured DNA). Cells were incubated with the probe for 30 min at 37 °C in a humid chamber and washed 3 times with 2× SSC containing 50% formamide for 20 min at 42 °C, and quickly washes 3 times with 2× SSC at RT. Next, immunostaining was performed as described above.

### 5.10. Cell Cycle Analysis

The U2OS and BJ-hTERT cells were pulsed with 30 µM BrdU ((Sigma Aldrich) for 30 min or 1 h and the DNA replication was followed after its withdrawal, respectively, for 0–6–9 or 12 h in cells transfected with scrambled siRNA or after silencing of POLL. The cells were then fixed in 1 mL of cold 70% ethanol and stored at 4 °C overnight. DNA was denatured with 2N HCl/0.5% Triton X-100 and neutralized with 1 mL of 0.1M sodium borate pH8.5. After centrifugation at 500g for 10 min, the cell pellet was resuspended with a mouse anti-BrdU antibody (1:100; Becton Dickinson,), diluted in 1% BSA, PBS 0.5% Tween 20, and incubated for 1 h at room temperature. Cells were then centrifuged at 500 g for 10 min and the pellet was resuspended with anti-mouse-488 (1:500; Alexa anti mouse 488, Molecular Probes/Invitrogen) and incubated for 1 h at room temperature. Samples were finally incubated with 10 ug/mL Propidium Iodide and analysed on a S3 flow cytometer (Bio-rad). Data were analysed using the program ProSort (Bio-rad) or FCS Express (DeNovo). 

### 5.11. CC-Assay

4 μg of genomic DNA were digested at 37 °C O/N with HinfI and RsaI (Thermo scientific) and RNase A (Sigma-Aldrich) in Buffer Tango 1x (Thermo scientific). 100 μL of digested DNA solution were subjected to isothermal amplification by incubation at 30 °C for 8h with 100 μL of Amplification Mix (0.2 mg/mL BSA, 0.1% Tween, 1 mM dNTPs) in the presence of ϕ29 Pol in its Reaction Buffer (Thermo scientific). Then DNA samples were spotted on a nylon transfer membrane (GeneScreen Hybridization Transfer Membrane, PerkinElmer), pre-hydrated in distilled H_2_O for 10′, using Bio-Dot Microfiltration Apparatus (Biorad). The membrane was then crosslinked on both sides with 254 nm UV radiation at 1200 J in order to covalently bind DNA to the membrane, washed 2 times in SSC2× for 5′ in agitation. Subsequently, the membrane was blocked in pre-hybridization buffer, pre-heated at 65 °C for 2 h (PerfectHyb Plus Hybridization Buffer, Sigma-Aldrich). The membrane was incubated with the C-rich 60-mer probe 5′-CCC TAA CCC TAA CCC TAA CCC TAA CCC TAA CCC TAA CCC TAA CCC TAA CCC TAA CCC TAA-3′ (biomers.net), 5′ labelled with T4 polynucleotide kinase (New England Biolabs) in the presence of [γ-^32^P]ATP, at 65 °C O/N. After washes, the membrane was dried and exposed to a phosphor screen for 24 h. The signal was detected by phosphorimaging (Typhoon, GE Healthcare).

### 5.12. Enzymatic Assays

For denaturing gel analysis of DNA synthesis products, the reaction mixtures contained 50 mM Tris-HCl (pH 7.0), 0.25 mg/mL BSA, 1 mM DTT, the dNTP’s as indicated in the figures and the indicated amounts of the 5′ ^32^P-labelled donor and the corresponding acceptor templates. Donor and acceptor templates were added at the beginning of the reaction along with the other reagents, without previous pre-incubation and/or annealing. A final Mg^++^ concentration of 1 mM was used. Concentrations of Pols, RP-A, POT1/TPP1 and dNTPs were as indicated in the respective Figures. Reactions were incubated 10 min at 37 °C, unless otherwise stated, and then stopped by addition of standard denaturing gel loading buffer (95% formamide, 10 mM EDTA, xylene cyanol and bromophenol blue), heated at 95 °C for 3 min and loaded on a 7 M urea/10% polyacrylamide gel.

### 5.13. Steady State Kinetic Analysis

Reactions were performed as described above. Quantification was done by scanning densitometry. The initial velocities of the reaction were calculated from the values of integrated gel band intensities: I*_T_/I_T-1_, where T is the target site, the template position of interest; I*_T_ is the sum of the integrated intensities at positions T, T + 1,…, T + *n*. All the intensity values were normalized to the total intensity of the corresponding lane to correct for differences in gel loading.

The apparent *K*_m_ and *k*_cat_ values were calculated by plotting the initial velocities in dependence of the 3′-OH concentrations (DNA) and fitting the data according to the Michaelis–Menten Equation:v = k_cat_[E]_0_/(1 + K_m_/[DNA])(1)
where [E]_0_, was the input enzyme concentration and k_cat_/[E]_0_ = V_max_. Substrate utilization efficiencies were defined as the V_max_/K_m_ ratio. Kinetic constants were derived fitting the experimental data to the appropriate equations with the computer program GraphPad Prism 3.0.

Dose-response curves for the inhibition of Pol l synthesis were fitted to the Equation:v = V_0_/(1 + [I]/IC_50_)(2)
where V_0_ is the apparent velocity of the reaction without the inhibitor, [I] is the variable inhibitor concentration and IC_50_ is the inhibitory potency, defined as the concentration of the inhibitor reducing the enzyme activity by 50%. Kinetic constants were derived fitting the experimental data to the appropriate equations with the computer program GraphPad Prism 3.0.

### 5.14. Statistical Analysis

As indicated in the Figure legends, mean values for each series of samples corresponding to the different conditions were analyzed by the computer program GraphPad Prism 3.0. Student’s *t* test was applied to paired samples. For Pol λ colocalization with telomeres and TPP1, statistical tests were applied combining all experiments from each series due to the limitations in numbers. The number of expected colocalizations was calculated as the product of the total dot positions in the cell area (i.e., calculated in pixel) and the combined frequency of dots obtained with different antibodies. The total number of possible dot positions in the cell was calculated as the number of dots contained in the cell area with a sliding area corresponding to a half of the dot size that represents the limit of visual resolution. Differences between the number of expected and observed colocalization were evaluated by the Chi-square goodness-of-fit test.

## Figures and Tables

**Figure 1 ijms-22-02365-f001:**
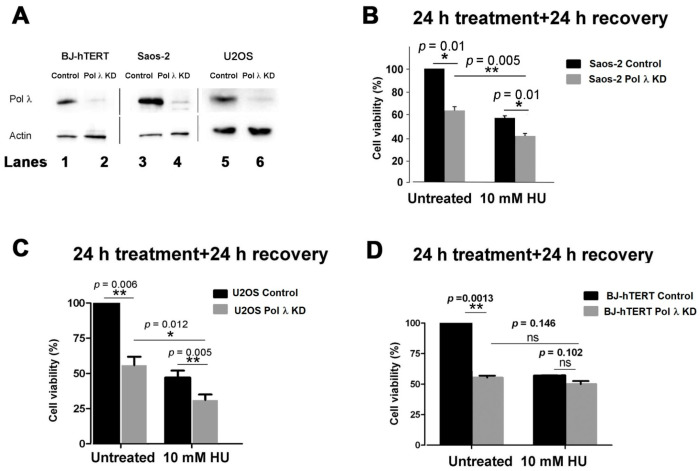
Silencing of human DNA polymerase λ specifically affects viability of ALT cell lines. (**A**) Western Blot analysis of Pol λ expression in BJ-hTERT, Saos-2 and U2OS cell lines either treated with scrambled siRNA control (lanes 1, 3, 5, respectively), or transfected with the anti-Pol λ siRNA si98 (lanes 2, 4) or the anti-Pol λ SMARTpool siRNAs cocktail (lane 6). (**B**) Viability of Saos-2 cells either treated with scrambled siRNA control (black bars) or silenced for Pol λ (grey bars) was measured by MTS assays. Cells were either untreated or treated with HU for 24 h and then viability was measured after an additional 24 h recovery in the absence of treatments. (**C**) As in panel B, but with the U2OS cell line. (**D**) As in panel B, but with the BJ-hTERT cell line. In all panels error bars represent ± SD of four independent experiments. *χ^2^ p* values are shown on top of the bars. n.s., not statistically significant (*χ^2^ p* value > 0.02). *, *χ^2^ p* value < 0.05. **, *χ^2^ p* value < 0.01.

**Figure 2 ijms-22-02365-f002:**
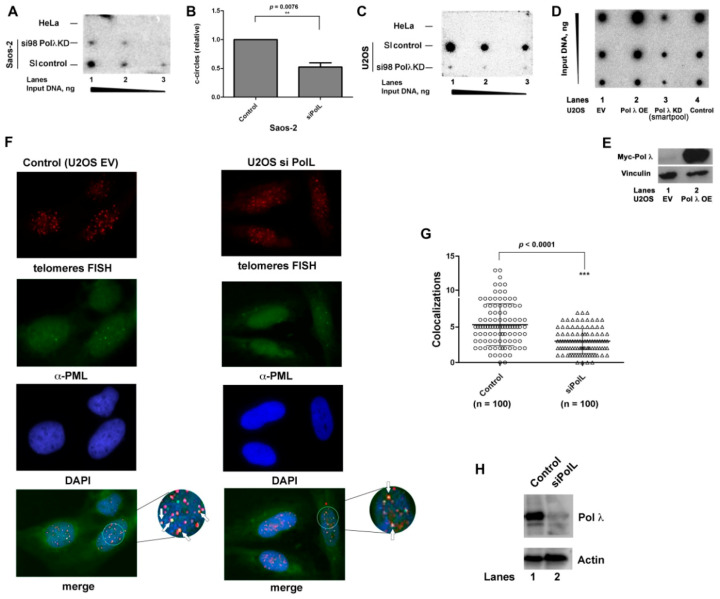
The levels of DNA polymerase λ correlate with the cellular ALT activity. (**A**) Dot blot assay for c-circle formation. Different amounts of amplified c-circle DNA from HeLa (top lane) or Saos-2 cells, either silenced for Pol λ (mid lane) or treated with scrambled control (bottom lane), were spotted on a membrane and hybridized with a radioactive telomeric probe. (**B**) Quantification of c-circles in SI control vs. Pol λ KD Saos-2 cells. Values are the mean of three independent experiments, each comprising three different amounts of DNA as in panel A. Error bars are ± S.D. The *χ^2^ p* value is shown on top. **, *χ^2^ p* value < 0.01. (**C**) As in panel A, but with c-circle DNA amplified from HeLa cells (top lane) or U2OS cells, either silenced for Pol λ (bottom lane) or treated with scrambled control (mid lane). (**D**) Dot blot assay for c-circle formation. Different amounts of amplified c-circle DNA from U2OS cells transfected with the empty vector (EV, lane 1), U2OS cells overexpressing Pol λ (OE, lane 2), U2OS cells silenced for Pol λ (KD, lane 3) or untreated U2OS cells (Control, lane 4), were spotted on a membrane and hybridized with a radioactive telomeric probe. (**E**) Western Blot analysis showing the levels of Pol λ in U2OS cells transfected with the Pol λ expression vector (corresponding to Pol λ OE of panel C). EV, U2OS cells transfected with the empty vector as negative control. (**F**) Confocal microscopy imaging of colocalization of PML protein (green) and telomeres (red) in U2OS cells either silenced for Pol λ (right panels) or control (left panels). Nuclei were stained with DAPI (blue). Representative colocalization events are indicated by white arrows. (**G**) Statistical analysis of three independent experiments comparing the number of colocalization events in control and silenced cells (n. of cells = 100 for each condition). The number of colocalization events observed within a single cell is shown for control (circles) and silenced (triangles) cells. *χ^2^ p* value is shown on top (***, *χ^2^ p* value < 0.0001). (**H**) Western Blot analysis showing the levels of Pol λ in U2OS cells either transfected with scramble control siRNA (lane 1) or anti-Pol λ siRNA (lane 2).

**Figure 3 ijms-22-02365-f003:**
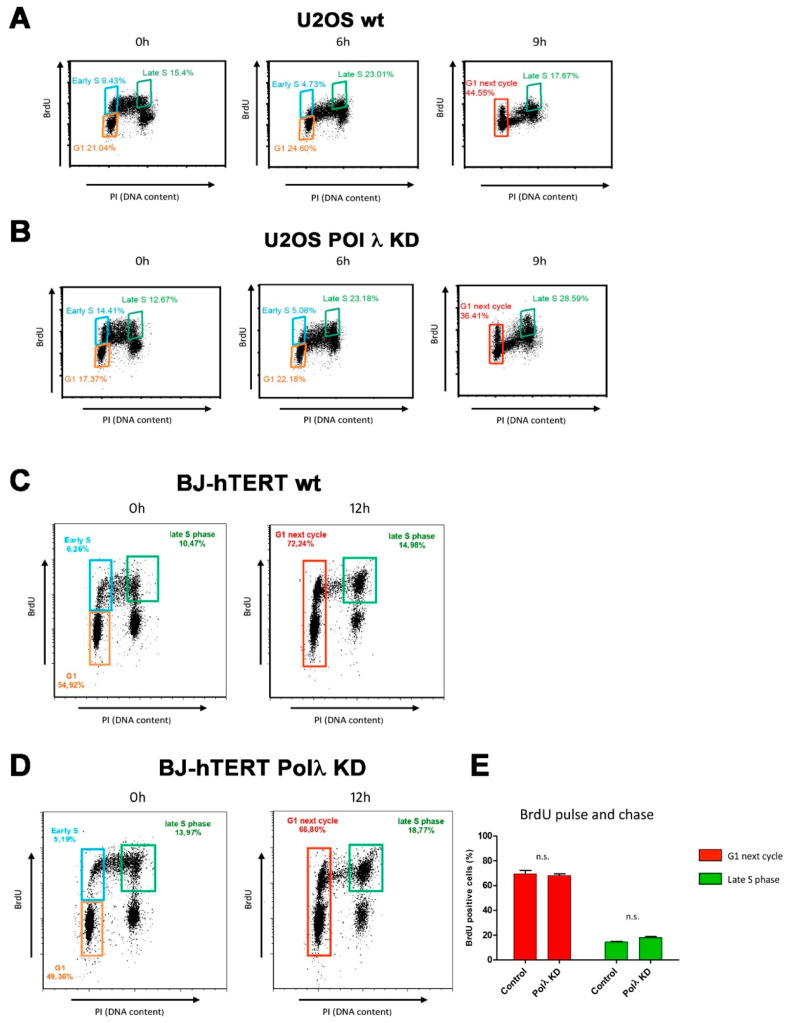
Pol λ silencing causes a slower progression through late S-G2 phases in ALT cells. The cell cycle progression was monitored by FACS. The orange gate indicates the percentage of cells in the G1-phase. The blue gate indicates the percentage of cells in early S-phase and the green gate indicates the late S-phase. The red gate at the 9 h time point indicates the early S-phase + G1-phase of the next cycle. (**A**) U2OS cells transfected with scrambled siRNA were pulsed with BrdU for 30 min and the DNA replication was followed after its withdrawal for 0–6–9 h in cells. The cells were stained with an anti-BrdU specific antibody and counterstained with propidium iodide (PI). (**B**) As in panel A but with U2OS cells silenced for Pol λ with the specific siRNA si98. (**C**) BJ-hTERT cells transfected with scrambled siRNA were pulsed with BrdU for 1 h and the DNA replication was followed after its withdrawal for 12 h in cells. The cells were stained with an anti-BrdU specific antibody and counterstained with propidium iodide (PI). (**D**) As in panel C but with BJ-hTERT cells silenced for Pol λ with the specific siRNA si98. (**E**) Quantification of cells in late S (green bars) or next G1 (red bars) for BJ-hTERT cells transfected with scrambled siRNA (Control) or silenced for Pol λ with the specific siRNA si98 (Pol λ KD). Values are the mean of three independent measurements. Error bars represent S.D. n.s., not statistically significant (Student’s *t*-test).

**Figure 4 ijms-22-02365-f004:**
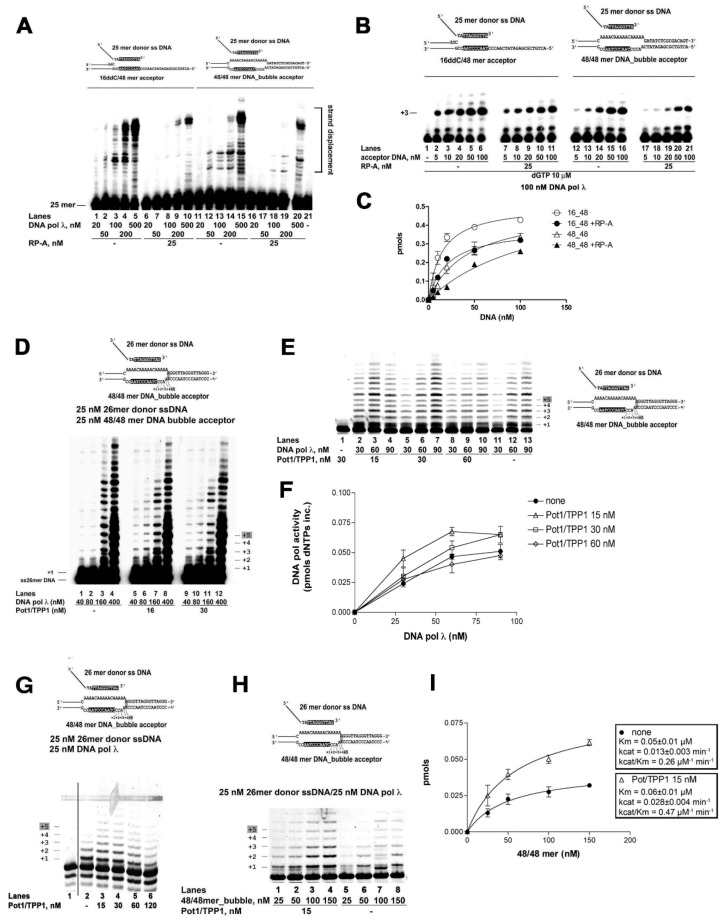
Microhomology-mediated strand transfer and DNA synthesis by DNA polymerase λ on telomere sequences is regulated by RP-A and POT1/TPP1. DNA templates are shown on top of each panel. Black boxes indicate the annealing site on the 5′-labelled primer and template strands. (**A**) Titration of Pol λ alone (lanes 1–5; 11–15) or in combination with 25 nM RP-A (lanes 6–10; 16–20), in the presence of the 16/48 linear (lanes 1–10) or the 48/48 “bubble” (lanes 11–20) acceptor DNA templates. (**B**) Titration of dideoxy-terminated 16/48mer (lanes 2–11) or 48/48mer “bubble” (lanes 12–21) acceptor DNA templates, in the presence of dGTP and in the absence (lanes 2–6; 12–16) or in the presence (lanes 7–11; 17–21) of 25 nM RP-A. (**C**) Dose-response curves of nucleotide incorporation (pmols) by Pol λ in the absence (clear symbols) or presence (filled symbols) of 25 nM RP-A on the linear (circles) or bubble (triangles) substrates illustrated in panel B. Values are the mean of two measurements ± S.E. (**D**) Titration of Pol λ on the 48/48mer “bubble” template, in the absence (lanes 1–4) or in the presence (lanes 5–12) of different amounts of the POT1/TPP1 complex. (**E**) Titration of Pol λ on the 48/48mer “bubble” template (25 nM), in the absence (lanes 11–13) or in the presence (lanes 1–10) of different amounts of the POT1/TPP1 complex. Lane 1: control in the absence of enzyme. (**F**) Quantification of nucleotide incorporation by increasing amounts of Pol λ in the absence (filled circles) or in the presence of increasing amounts of POT1/PP1 as indicated in panel E. Values are the mean of two measurements ±S.E. (**G**) Pol λ activity on the 48/48 mer acceptor template in the absence (lane 2) or in the presence (lanes 3–6) of increasing amounts of POT1/TPP1. (**H**) Titration of the 48/48 mer acceptor template in the presence of Pol λ and in the presence (lanes 1–5) or in the absence (lanes 6–9) of POT1/TPP1. (**I**) Dose-response curves of nucleotide incorporation (pmols) by Pol λ in the absence (clear symbols) or presence (filled symbols) of 15 nM POT1/TTP1 on the bubble substrate shown in panel H. Values are the mean of two measurements ± S.E.

**Figure 5 ijms-22-02365-f005:**
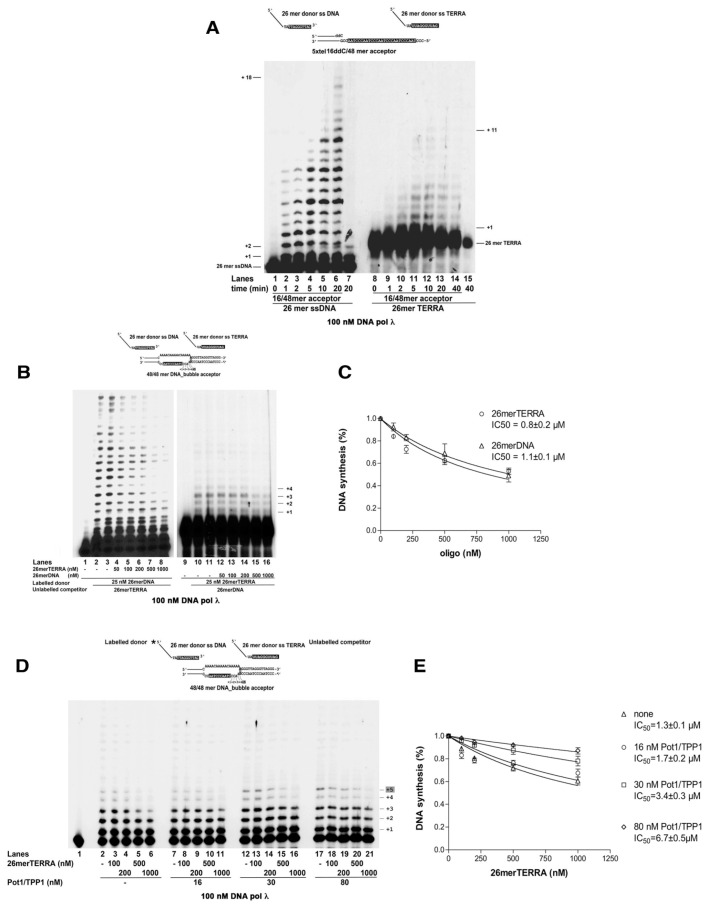
TERRA RNA represses the strand transfer activity by DNA polymerase λ and is counteracted by POT1/TPP1. (**A**) Time course of MMST synthesis by Pol λ with the 26 mer donor DNA (lanes 1–6) or the 26 mer TERRA donor RNA (lanes 8–14) and the 5xtel 16/48 mer acceptor template. Lane 7, reaction with the 26 mer donor DNA without the acceptor template. Lane 15, reaction with the 26 mer TERRA donor RNA without the acceptor template. (**B**) Titration of 26mer TERRA (lanes 2–8) or the 26mer DNA (lanes 9–16) in the DNA-dependent (lanes 2–8) or TERRA-dependent (lanes 9–16) MMST reaction catalyzed by Pol λ on the 48/48mer bubble acceptor template. (**C**) Dose response curves of the inhibition of Pol λ synthesis by the competitor 26mer TERRA (circles) or 26mer DNA (triangles) shown in panel B. Values are the mean of two measurements ± S.E. (**D**) Titration of 26mer TERRA in the DNA-dependent MMST reaction catalyzed by Pol λ on the 48/48mer bubble acceptor template, in the absence (lanes 2–6) or presence (lanes 7–21) of increasing POT1/TPP1 concentrations. (**E**) Dose response curves of the inhibition of Pol λ synthesis by the competitor 26mer TERRA, in the absence (triangles) or in the presence of increasing amounts of POT1/TPP1. Values are the mean of two measurements ± S.E.

**Figure 6 ijms-22-02365-f006:**
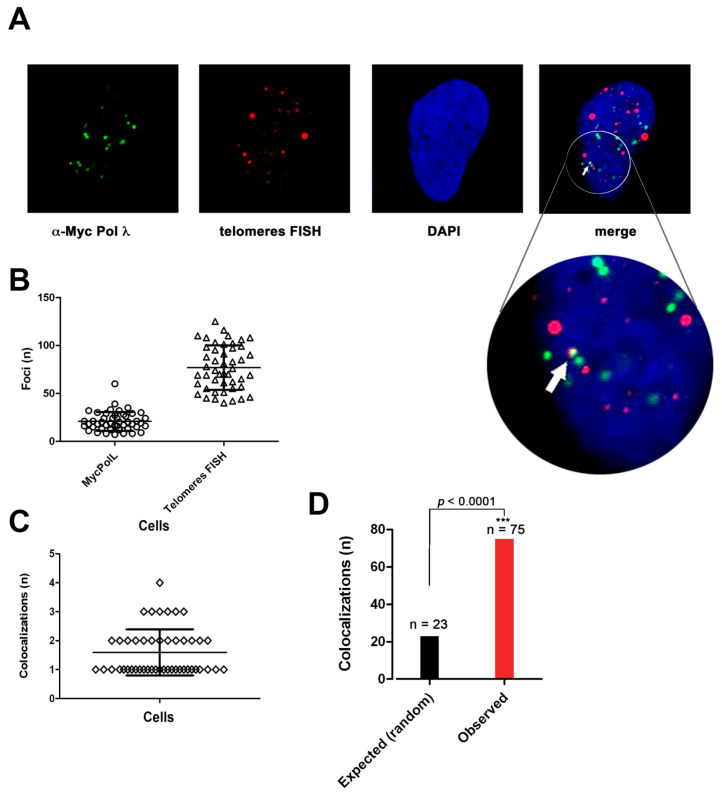
DNA polymerase λ associates with telomeres in ALT cells. (**A**) Confocal microscopy imaging of Pol λ colocalization with telomeres in Saos-2 λ13 cell clone stably expressing c-Myc Pol λ. A representative colocalization event is indicated by the white arrow. (**B**) The number of signals (foci) detected, respectively, for Pol λ (circles) and telomeres (triangles) within a single cell is shown. (**C**) Number of observed colocalization events per cell. (**D**) Statistical analysis comparing the expected number (n) of colocalization events under the hypothesis of random chance colocalization, with the number of observed colocalizations. χ^2^
*p* value is shown on top of the bars. ***, χ^2^
*p* value < 0.005.

**Figure 7 ijms-22-02365-f007:**
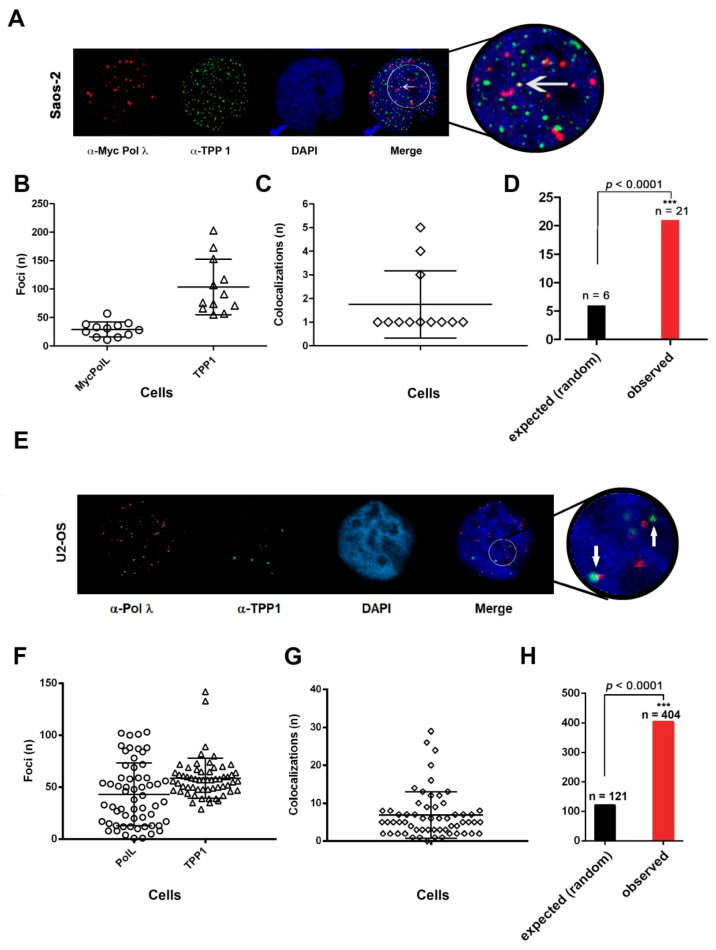
DNA polymerase λ associates with TPP1 in ALT cells (A-D Saos-2; E-H U2OS). (**A**) Confocal microscopy imaging of Pol λ colocalization with TPP1 in Saos-2 cells stably expressing c-Myc Pol λ. The white arrow indicates a representative colocalization event. (**B**) Number of foci detected for Myc-Pol λ (circles) and TPP1 (tringles) per cell. (**C**) Number of observed colocalization events per cell. (**D**) Statistical analysis comparing the expected number (n) of Pol λ-TPP1 colocalization events under the hypothesis of random chance colocalization, with the number of observed colocalizations. χ^2^
*p* values are shown on top of the bars. (**E**) Confocal microscopy imaging of Pol λ colocalization with TPP1 in U2OS cells. The white arrow indicates representative colocalization events in a single stack. (**F**) Number of observed foci for Pol λ (circles) and TPP1 (triangles) per cell. (**G**) Number of observed colocalization events per cell. (**H**) Statistical analysis comparing the expected number (n) of Pol λ-TPP1 colocalization events under the hypothesis of random chance colocalization, with the number of observed colocalizations. χ^2^
*p* values are shown on top of the bars. ***, χ^2^
*p* value < 0.005.

**Figure 8 ijms-22-02365-f008:**
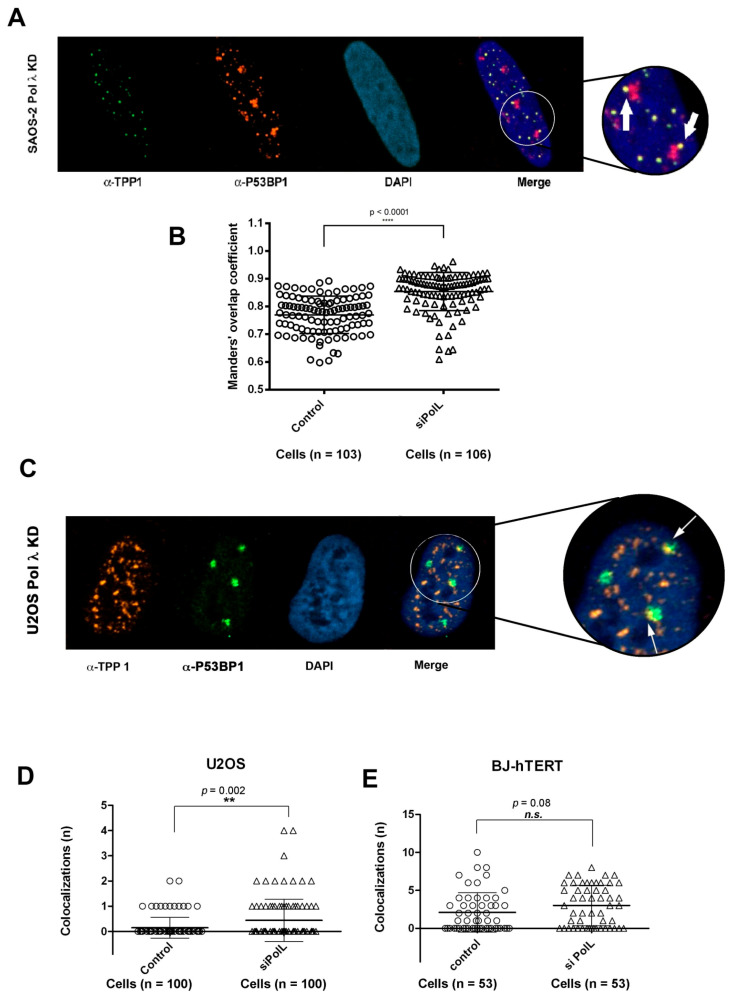
DNA polymerase λ silencing causes telomere stress specifically in ALT cells. (**A**) Confocal microscopy imaging of TPP1 colocalization with P53BP1 (telomere dysfunction foci, TIF) in Saos-2 cells silenced for Pol λ. The white arrows indicate representative colocalization events (TIF). (**B**) Quantification of TPP1-P53BP1 colocalization events through JACoP plug-in of ImageJ. 10^3^ and 10^6^ cells were analyzed, respectively, for cells treated with scramble control siRNA (Control) and for cells silenced for Pol λ (siPolL). Each circle and triangle represents the amount of colocalizing signal within a single cell. These data are quantified by M2 Mander’s overlap coefficients, that are 0.7702 (SD = 0.06734) for cells treated with scrambled control siRNA and 0.8537 (SD = 0.06885) for cells silenced for Pol λ. Error bars represent ±SD. *p*-values were calculated through two-tailed Student’s *t*-test (95% confidence interval) (**C**) Confocal microscopy imaging of TPP1 colocalization with P53BP1 (telomere dysfunction foci, TIF) in U2OS cells silenced for Pol λ. The white arrows indicate representative colocalization events (TIF). (**D**) Statistical analysis comparing the number of TPP1-P53BP1 colocalization events in U2OS treated with scrambled control siRNA (control) or treated with anti-Pol λ siRNA (siPolL). n. of analyzed cells = 100 for each condition. χ^2^
*p* value is shown on top of the bars. (**E**) Statistical analysis comparing the number of TPP1-P53BP1 colocalization events in BJ-hTERT cells treated with scrambled control siRNA (control) or treated with anti-Pol λ siRNA (siPolL). n. of analyzed cells = 53 for each condition. χ^2^
*p* value is shown on top of the bars. n.s., not significant, χ^2^
*p* > 0.02. ** *p* value = 0.002, **** *p* value < 0.0001.

**Table 1 ijms-22-02365-t001:** Effect of RP-A on the kinetic parameters for template binding by DNA polymerase λ during MMST DNA synthesis.

Acceptor Template	16/48		16/48 5xtel	48/48 Bubble	
	−RP-A	+RP-A	+RP-A	−RP-A	+RP-A
K_m_ (nM)	12 ± 1	21 ± 2	11 ± 1	43 ± 4	124 ± 10
V_max_ (nM × min^−1^)	9 ± 1	7 ± 1	7.5 ± 0.7	9 ± 1	10 ± 1
V_max_/K_m_ (min^−1^)	0.75	0.33	0.68	0.21	0.08
-fold reduction V_max_/K_m_		2.27	1.1		2.62

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
