# Peer review of "A Role for Human DNA Polymerase λ in Alternative Lengthening of Telomeres"

_ijms, 2021, doi:10.3390/ijms22052365_

Round 1

Reviewer 1 Report

This study by Mentegari, Bertoletti, Kissova and colleagues is very well conducted and significant for the ALT field.

I have only some questions and suggestions to improve some of the data.

In figure 1, The authors describe a defect of cell growth upon pol L depletion even when untreated. Are these cells dying by apoptosis or are they just slower in the cell cycle as would suggest sup figure 2?

I am also wondering why this sup figure 2 is not included in the main figures as it seems a significant data to show.

Figure 2 indicates that PolL promotes ALT features. What about T-SCEs?

Figure 3 indicates some colocalization between the telomeres and exogenous PolL. Are these cell synchronized? Would this colocalization increase if cells were in late S phase as suggested by the sentence 169-171 page4? This figure seems to complement the data presented in figure 6 and therefore could many be merged as I am not sure how FISH staining brings a different information than TPP1 staining.

In Figure 4, quantification with replicates seem to be missing. These gels are obtained by radioactivity and could therefore be quantified using Image quant to obtain the % of elongation and/or the processivity/activity of the enzyme. Some statistical analysis rather than visual appreciation would bring more strength to the conclusions of the authors.

Finally, did the authors look at the telomeres length in PolL depleted cells? If PolL is promoting ALT, then telomeres in PolL cells should be shorter. This could be performed using TeSLA to have accurate telomere sizes.

Reviewer 2 Report

I thoroughly enjoyed reading this article. The questions asked were relevant and the methods used to test them were thoughtful. The quality of figures were of high standards. Overall this is a high quality manuscript.

The only experiment I would ask the authors is to overexpress human polymerase  λ in the knockdown cells (Fig 1) and test whether it rescues the viability of the cells to control levels.  

Round 2

Reviewer 1 Report

I thank the authors for their thorough response. I enjoyed reading this new version of the manuscript.